# A dual mode self-test for a stand alone AES core

**Fakir Sharif Hossain**[1]*, **Taiyeb Hasan Sakib**[2], **Muhammad Ashar**[3], **Rian Ferdian**[4]

**1** Department of Electrical and Electronic Engineering, Ahsanullah University of Science and Technology, Dhaka, BD, **2** Department of Electrical and Electronic Engineering, Brac University, Dhaka, BD, **3** PUIPT Disruptive Learning Innovation, University Negeri Malang, Malang, Indonesia, **4** Faculty of Information Technology, University of Andalas, Dedung Rektorat, Padang, Indonesia

☉ These authors contributed equally to this work.

\* dr.hossain@baiust.edu.bd

**Data Availability Statement:** All relevant data are within the paper.

**Funding:** Unfunded studies Enter: The author(s) received no specific funding for this work.

## Abstract

Advanced Encryption Standard (AES) is the most secured ciphertext algorithm that is unbreakable in a software platform's reasonable time. AES has been proved to be the most robust symmetric encryption algorithm declared by the USA Government. Its hardware implementation offers much higher speed and physical security than that of its software implementation. The testability and hardware Trojans are two significant concerns that make the AES chip complex and vulnerable. The problem of testability in the complex AES chip is not addressed yet, and also, the hardware Trojan insertion into the chip may be a significant security threat by leaking information to the intruder. The proposed method is a dual-mode self-test architecture that can detect the hardware Trojans at the manufacturing test and perform an online parametric test to identify parametric chip defects. This work contributes to partitioning the AES circuit into small blocks and comparing adjacent blocks to ensure self-referencing. The detection accuracy is sharpened by a comparative power ratio threshold, determined by process variations and the accuracy of the built-in current sensors. This architecture can reduce the delay, power consumption, and area overhead compared to other works.

## 1 Introduction

Advanced Encryption Standard, the short form is AES, is the world's most secured cryptographic algorithm established by Rijmen and Demon and got approval by NIST, US in 2001 [1]. Numerous crypt-analytical attacks such as brute-force, Linear crypt-analysis, differential crypt-analysis, etc., are failed to display a potential threat for the security of the AES [2]. For example, a brute force attack is a potential threat to retrieve the plaintext by reading the number of rounds in AES [3]. The AES-128 cryptography system provides an approximated time frame of one billion years to mount a brute force attack to retrieve the plane text. Due to this impressive security potentiality of AES, it is being used in various emerging applications, either in software or hardware implementations. Hardware implementation of the algorithm offers higher security and speed than that of its software implementation. Due to enormous speed

**Competing interests:** NO authors have competing interests Enter: The authors have declared that no competing interests exist.

and security performances, now a lot of research for hardware realization of the AES crypto-processor is reported in the literature [4–13]. Some of the researches focus on hardware resource optimization [4–6], while some other on speed optimization [7–9] and some other on power consumption optimization [10–13]. A very few works on built-in-self-test (BIST) has reported in literature [14–18]. Some of them focus on-chip test pattern generation on detecting circuit aging [14–16], and some other structures detect Trojans [17, 18]. The testability and hardware Trojans are two major concerns that make the AES chip complex and vulnerable. The problem of testability in the complex AES chip is not addressed yet, and also, the hardware Trojan is a significant security threat that can leak secret key information easily if the hardware is compromised. Hardware Trojans are the manipulation or insertion of some extra transistors in a chip which can result in information leaking from the AES chip [19]. The research motivation of this work is to facilitate the BIST implementation in AES cryptography processors in terms of testability and hardware security domain. From a testability perspective, the on-chip BIST structure can significantly reduce the test cost with extra hardware and performance overhead. There is a great concern of hardware-based attacks from the security domain, like hardware Trojans in recent state-of-the-art. The AES chip is vulnerable to hardware Trojan attacks, as some significant research displayed. Therefore, the main purpose of this research is to address these problems by implementing a dual-mode BIST technique into the AES chip to address the testability issues and analyze the overhead of area, delay, and power.

Testing chips is a rigorous process and very costly due to Automatic Test Equipment (ATE) involvement. For years chip manufactures have been complaining about the increasing testing cost against ATE vendors [20]. Therefore, researchers are trying to find an alternative solution to reduce the test cost. The System-on chips (SoC) is one of the optimized solutions reducing the test cost. Software-based self-test is one kind of SoC testing that facilitates testing on-chip [21]. On-chip testing is becoming a widespread practice for the chip designer to reduce the test cost by eliminating the use of ATE. If a chip itself has embedded test equipment, it can reduce the cost of expensive testing externally. Software-based self-test (SBST) is an on-chip testing facility to test the chip with built-in test devices and also facilitates the advantages of online tests [21–24]. A chip may fall in defects in operation, or its performance can be deteriorated or aged with time going on. SBST architecture can perform a routine test to deliver the healthy chip with its internal circuitry.

Numerous SBST structures are in practice in many industrial chips [25–30]. In this work, we propose a dual-mode self-test (DMST) architecture to test the AES cryptography chip in two phases: hardware Trojans test at the post-manufacturing stage and routine fault testing after deployment chip in the device. It is the first attempt to propose a stand-alone AES core that can detect both faults and Trojans to the best of our knowledge. Our proposed approach facilitates identifying Trojans in manufacturing tests and identifies faults while the chip is in operation.

Proposed DMST rests on a power side-channel analysis-based self-referencing architecture that can identify manufacturing defects and Trojans. Circuit partitioning, adjacent blocks comparison, process variations, and environment noises are considered design-for-security states. We choose power side-channel over logic testing as the logic testing approaches require a particular pattern sequence to identify the effect of logic failure. Generating such a test vector set is challenging since the number of combinations is very high [31–33]. On the other hand, there is an absolute advantage of using side-channel analysis. Faults or Trojans' presence inside an integrated circuit (IC) causes specific distortion, affecting side-channel parameters like power, delay, and electromagnetic wave. Analyzing these effects can identify defects or malicious effects where complete triggering is not required. However, an excessive number of

noises like process variations, sensor accuracy, environmental noise, etc., prevents identification. Among these, process variation is particularly a severe problem and a natural phenomenon that unavoidably arises during the manufacturing process of ICs. Due to process variations, transistor parameters deviate from their nominal values, resulting in false-positive outcomes.

To deal with the process variations, we propose a comparative power ratio (CPR) threshold, determined by process variations and sensor accuracy of our proposed built-in current sensors (BICS). It may not be possible to set the threshold value as null as there always are noises that exist. The *CPR* threshold parameter is considered to identify these noises. The proposed DMST approach works by partitioning the AES-128 circuit into small blocks, activating each block by several selected transition delay test patterns (TDPs), and meeting an equal power consumption to adjacent blocks to ensure self-referencing. Self-referencing is ensured by identifying a pair of test patterns denoted as PEAB (power-equal adjacent block) pairs that consume equal power in their respective adjacent blocks. The proposed DMST method needs to overcome several challenges such as test set, detection particulars, process variations, sensor accuracy, area overhead, and power consumption.

The concept of partitioning a circuit builds on our previous approach [34] that shows a more realistic clock tree-driven partitioning technique. A clock tree-defined circuit partitioning ensures a minimal hardware overhead to deliver self-referencing to eliminate the inter-chip variation effect. An augmented pattern set that provides n-detection and virtual observation point (VOP) insertion with adjacent block referencing delivers reduced intra-chip systematic and random variation effects and increases fault coverage. The proposed architecture is based on a scan-based self-test where test patterns are applied through scan chains to activate each block individually to measure the power value. Each time a single block gets activated and the remaining blocks keep frozen, the power consumption is reduced significantly. The proposed DMST architecture consists of design-for-testability and design-for-security features. The contributions of this work are:

- to introduce the concept of dual-mode built-in-self-test technique in designing the AES Crypto-processor chip for detecting Trojans and manufacturing faults. The idea is new and not reported in the literature yet.

- parametric measurement performed by embedding current sensors near multiple power rails to determine the power consumption by a partition's activated cells individually. power consumption optimization by activating a single partition while keeping others frozen.

- variation aware equal power consuming partitions pair to minimize the effect of on-chip process variations.

- transition delay fault patterns calibrated with virtual test point insertion to deliver 100% fault coverage.

- proposal of two testing modes: the first is in the test mode to identify Trojans once after fabrication, and the second is for the online self-test mode through scan chains.

- optimized design to deliver minimal hardware overhead and less power consumption.

The remaining sections of this paper are presented in the following manner. Next Section-2 describes the AES algorithm and a general DMST architecture. A relation between process variation and power is considered with a literature review on fault identification with the Trojan detection technique. The proposed DMST architecture is presented in Section 3, and experimental results are shown in Section 4. Finally, the work is concluded in Section 5.

## 2 AES DMST and process variation

### 2.1 AES algorithm

Vincent Rijmen and Joen Daemen developed an advanced Encryption Standard (AES) or Rijndael algorithm. The algorithm was a derivation of the block cipher family. In 2001, the Rijndael algorithm was submitted as an entry to the National Institute of Standards and Technology's (NIST) competition to select an Advanced Encryption Standard (AES) to replace Data Encryption Standard (DES). Rijndael won the battle, and NIST officially selected the 128, 192, and 256-bit versions of Rijndael as the Advanced Encryption Standard.

The AES has two main processes, the encryption process, and the cipher key scheduling process. The encryption has four sub-processes, which are subByte, shiftRows, mixColumns, and addRoundKey. The subByte substitutes each byte from the input data into a new byte from a pre-generate lookup table. Furthermore, shiftRows cyclically shift the last three rows of the previous step state in a certain number of shifts. The mixColumns then combines every four of the previous state data by utilizing a Galois Field Matrix multiplication. The addRoundKeys performs a bitwise XOR process between the data state and the obtained round keys.

The second process in the AES is scheduling the chipper key by performing an XOR operation between the current state of the cipher key with round constants (rCon). In AES, the cipher key is an array of 32 bits. The round constant is obtained from the first column of the current state cipher key array. This column is shift rotated then non-linear transformed in subByte operation. However, the AES still considered having a detailed Keyschedule and basic encryption operations. Many attacks are stringent upon this property. On the mentioned bit versions of the AES, 128-bit is focused on this work. Understanding this version may provide the necessary background to understand other versions as well.

### 2.2 AES DMST

A built-in self-test or BIST is a structural test method that adds logic to an IC, which allows the IC to test its operation periodically. BIST allows ICs to test themselves with onboard testers. In conventional BIST, one defect in the circuit can affect thousands of test vector outputs. Therefore, the main challenge is to detect the exact fault location in the IC from thousands of signature test responses. The dual-mode self-test (DMST) is a promising method to overcome the fault detecting problem. DMST is a technique to detect the fault location by acquiring direct access to the circuit under test (CUT) nodes. The DMST technique can guarantee correct fault detection regardless of the number of errors in the test output vectors.

DMST localizes the Trojan detection into several blocks in an IC. Implementing DMST can minimize the complexity and improve the quality of translating the test vector, resulting in detecting the circuit irregularity. However, the DMST requires additional flip-flops to gain direct access to the IC blocks. The additional sub-circuit also affects the whole system area and power. Thus, the researchers have provided several results on reducing the number of flip-flops elements.

The general DMST structure can be seen in Fig 1. The linear feedback shift register (LFSR) is utilized to create signature input patterns for the CUT. The signature pattern here is explicitly designed to detect each node defect. Then N nodes in the CUT are directly connected to the scan flip-flop to record the response.

### 2.3 Process variation in power side-channel

With the rapid growth of transistor scaling, modern ICs often suffer from elevated process variations. Side-channel parameters experiencing process variations always cause the

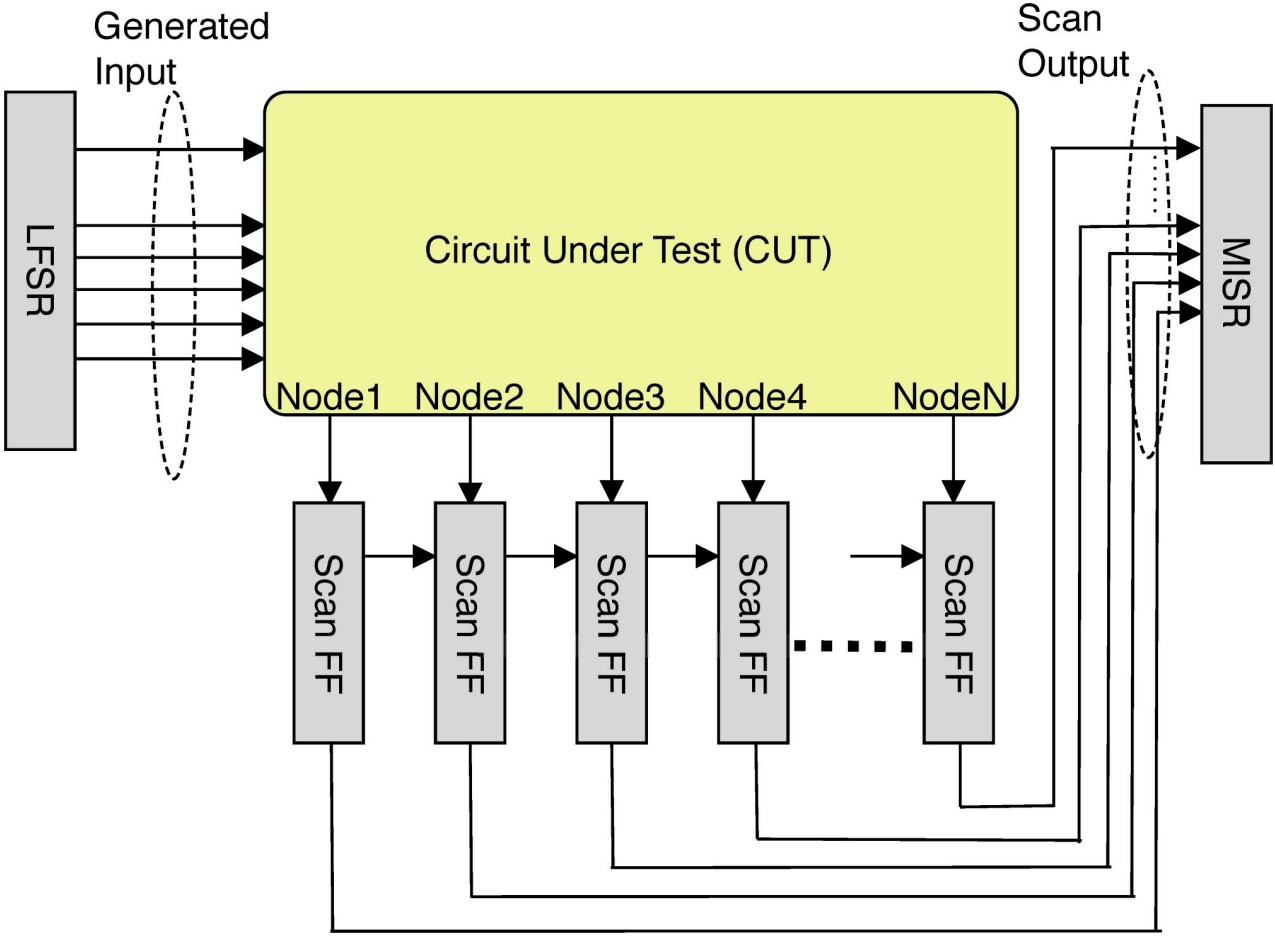

**Fig 1. A general dual-mode self-test (DMST) structure.**

fundamental IC properties to diverge from their nominal design specifications in random and systematic manners. Process variations are categorized into inter-chip systematic variation, intra-chip systematic variation, and random variation, and these affect side-channel signatures, including power and delay [35].

It is reported that process variation results in up to twenty times variation in standby leakage and around 30% in chip frequency in 180nm technology [36]. The breakdown of process variations in ring oscillator frequencies for 65nm technology is reported to be systematic 70% of inter-chip, 20% of intra-chip, and random 10% variations [37]. An experiment in real IC for 130nm technology showed that two cells within 1 or 2 m distance have a strong correlation in critical dimension. In contrast, cells have a low correlation if they are separated by over 10m [38].

There are two power sources, dynamic power, and leakage power. Leakage power is an important design consideration as it exponentially increases depending on device parameters, such as threshold voltage. Leakage powers from different chips distribute in a wide range of variation windows caused by inter-chip variation as reported in [39]. Though leakage power also depends on patterns, the same chip's leakage powers keep the same trend of inter-die variation, and the variation window for one chip is much narrower than the variation window for different chips. Regarding dynamic power, it involves only switching cells. Inter-chip and

intra-chip systematic variations affect dynamic power when more cells get switching. Oppositely, the standard deviation caused by random variations becomes small if more cells switch concurrently. When $n$ cells switches, the standard deviation becomes $\sigma/\sqrt{n}$ for the standard variation $\sigma$ of one cell. Therefore, if a sufficient number of cells switches, their random variation effects can cancel each other. However, it is the case where the same set of switching occurs in different chips. When multiple cells are switching, the more cells are switching, the more dynamic power is affected by variation if the cells have similar variation trends.

## 2.4 Related works

**2.4.1 Parametric fault detection.** Parametric fault detection is conveniently used for circuit testing to determine components' catastrophic faults and fractional deviations from their nominal values. Traditional techniques of fault detection require checking performance parameters against tolerance ranges given by the designers [40]. These techniques take complex computation and significant time. Therefore, researchers proposed various parametric tests for fault detection among which mentionable are transfer function coefficients [41], input current waveform [42], network parameters [43] and polynomial coefficients of input voltage [44].

The probabilistic nature of faults under process variations is considered in parametric fault simulation in [45]. Parametric faults are mainly due to poor manufacturing control. For high-frequency systems, diagnostics for analog and digital parts require different methods. Hence, making the testability is difficult and limited. Structural testing is deployed to solve this problem [46]. The minimum node selection method to increase parametric fault detection efficiency is discussed in [47].

**2.4.2 Power side-channel based hardware Trojan detection.** Up until 2021, the presence of Hardware Trojans did not be detected in practical products [48]. However, researchers have defined several techniques to determine hardware Trojans to ensure future chip security. Detection of hardware Trojans is susceptible and requires testing before marketing. Power side-channel analysis is one of the potential detection techniques among numerous techniques. Trojan detection with IC fingerprints using power or temperature is earlier works [49]. Power consumption of the device under testing is measured and compared with a golden device to detect Trojans [48]. A related experiment was carried out with FPGA and oscilloscopes to detect Trojan employing power fluctuation [50]. Trojans are introduced below the noise power level of the device, and detection is based on power consumption level [51]. Power-based side-channel attack is studied in [52].

## 2.5 Self-test based fault and Trojan detection

DMST is a technique to localize the Trojan detection into several blocks in the AES chip. The implementation of DMST can minimize the complexity and improve the quality of translating the test vector results to detect the circuit's irregularity. However, the DMST requires additional flip-flops to gain direct access to the AES partition. The additional sub-circuit also affects the whole system area and power. Thus, the researchers have provided several results on reducing the number of flip-flops elements [53].

Further development in the test vector set of DMST with pseudo-random numbers reduces the test's deterministic pattern. It enables a lower number of flip-flops while improving the detection quality [54]. The compression technique was also introduced to lower the area of the required sub-circuit [55, 56]. There are some other proposals for reducing the energy requirement of the DMST circuit. The lower energy consumption in the test sub-circuit can be

obtained by applying a clock disabling mechanism [57]. The DMST technique improvement also includes methods to reduce the test points and the testing time [58].

## 3 Proposed DMST architecture

A self-referencing method is proposed utilizing circuit partitioning and selecting variation-aware test patterns to measure the current of a single block and stored in a memory to compare with neighboring blocks. A clock tree-based circuit partitioning and transition-delay patterns are the main components for the current sensors to measure power values. The proposed power equal adjacent block (PEAB) pair consumes similar power to the neighboring partitions or blocks. Each block is activated through test patterns, and a gating clock controller freezes other blocks of the chip.

A variation-aware transition delay test pattern selection process is proposed to obtain Trojans and parametric defect-free AES chips. The test patterns set is variation-aware because a comparison of two adjacent blocks eliminates inter-chip and reduces intra-chip process variation effects. If a sufficient number of cells in a block can be activated applying test patterns, a reduced effect of intra-chip random variation can be achieved. PEAB pairs ensure a sufficient number of cells toggling to diminish intra-chip random variation effects.

### 3.1 Fault and threat models

Fault injection is a testing purpose to learn about system behavior under unusual stress on the system. Fault injection can perform virtually or in a real chip. The proposed method injects fault by changing the hardware in a chip, called hardware fault injection. In the case of hardware Trojans, it takes extra hardware inserting in a chip, which may consume excessive power consumption if activated. Hardware fault injection is introduced in two ways; hardware fault injection with contact and without contact as reported in [59]. This paper considers only internal fault injection and extra hardware insertion to evaluate the proposed approach.

A fault model is a group of faults where it considers various faults like permanent fault like stuck-at-fault, transition delay fault, aging fault, temperature fault, and some others. The assumption is that a fault may change the parametric value like power consumption of the faulty block of a chip and show deviation of power value compared to two adjacent blocks. Similarly, the proposed threat model is to insert some extra logical hardware into some blocks of a chip, and a power difference of two blocks may result in Trojan identification. It acknowledges that the common activation of a Trojan by two adjacent blocks may fail to deliver sufficient power, failing to detect.

The hardware Trojans are detected with self-referencing using PEAB pairs; however, it may be possible that the PEAB pairs can be created with Trojans already included. In this case, it may not be possible to detect the Trojans through power consumption measurement, although the parametric defect detection through self-test using the proposed architecture can be done normally. Even if the PEAB pairs are created through the simulation, it is not easy to guarantee that the simulation is performed on a Trojan-free golden circuit. Therefore, we consider the AES-128 circuit netlist from TrustHub [60] as a trusted and golden netlist. All the simulations are performed on the golden netlist of the AES-128.

### 3.2 Overview

Parametric defects and Trojans are detected in power equal adjacent block self-referencing approach if a sufficient power difference can be measured in two adjacent blocks. As the PEAB pairs are power equal, an adequate difference in power values can identify parametric defects or Trojan inserted. Although there is no defect or Trojan in the chip, it may have false positive

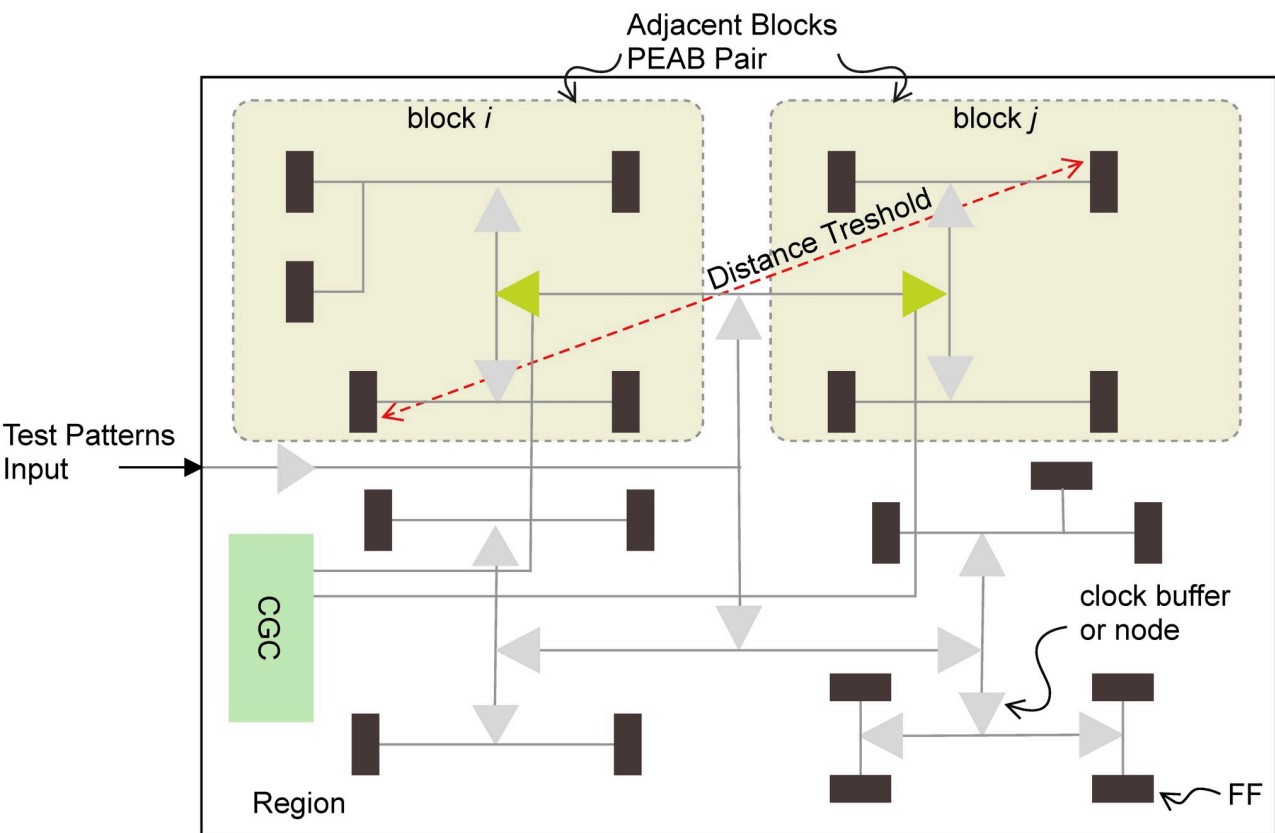

**Fig 2. Definition of nodes, adjacent blocks, region, distance threshold and PEAB pairs.**

detection because of a significant power difference. This false positive is due to process variations. The proposed method identified this level of process variation and termed it as a variation threshold. The variation threshold ensures the minimum power difference value from the process variation below which PEAB pairs cannot identify the chip as faulty. Process characteristics set the threshold.

The proposed approach signifies the idea of circuit partitioning utilizing the clock tree of the circuit. We select a set of Launch-on-Capture (LOC) transition patterns so that it delivers power equal blocks in a pair to perform self-referencing. For a clear understanding, some terminologies are provided that are used throughout this paper.

- Node: A node is a clock buffer in the clock tree. There are 50 end nodes in the AES circuit.

- Block: A group of sequential elements or flip-flops connected to a clock node, see Fig 2.

- Region: A set of cells activated by one block and a test pattern. A block is said to be a region if a set of both combinational and sequential cells are activated simultaneously by test patterns through scan chains. The scan flip-flops create a virtual boundary of activated cells after applying a test pattern. Therefore, a set of test patterns applying to a single node while keeping other nodes gated delivers a virtual boundary for the node, which is termed as a region. Activating a single region by test patterns follows a one-hot criterion, thus activating one block per pattern while keeping remaining blocks frozen see Fig 2.

- Adjacent blocks: Two blocks are adjacent if their sequential elements are located within a distance threshold. The distance threshold is measured by the geometric distance of flip-

flops in the chip layout. The distance threshold value is kept sufficiently low so that the difference of dynamic power consumption due to the intra-chip process variation diminished significantly see Fig 2.

- PEAB pair: Equal Power consuming test pattern pairs defined by a block ID and test pattern specification for each. The PEAB pair is a pair of tuples of a block and a test pattern. Let $B_i$ and $B_j$ be two blocks, and $t_i$ and $t_j$ be test patterns. A couple of two tuples $(B_i, t_i)$ and $(B_j, t_j)$ is a PEAB pair whose simulated power values when $t_i$ is applied to $B_i$ and when $t_j$ is applied to $B_j$ are almost identical see Fig 2.

The proposed PEAB pair is generated from transition delay patterns in launch-on-capture mode by the ATPG tool. Patterns are selected by inserting virtual observation points (VOP)s in the circuit's netlist, and 100% toggling is ensured through $n$-detect test. Selected PEAB pairs deliver numerous benefits. A PEAB pair compares two blocks in a chip. It eliminates the inter-chip variation effect, and a strong spatial correlation between adjacent blocks reduces the intra-chip systematic variation effect. Activation of a sufficient number of cells facilitates a reduced intra-chip random variation. These considerations can deliver a reduced variation threshold, thus significantly identifying defects and Trojans.

PEAB pair generation meets a sensitive detection under various noises. Fig 3 shows an overview of the proposed method. Circuit blocks are generated by selecting clock buffer or node in the clock [34]. PEAB pairs are generated through partitioned blocks and test patterns. Initial partitioned blocks are the primary set of pairs. This preliminary set is selected as the final one if it delivers sufficient toggling. Insufficient toggling is improved by inserting VOPs by ensuring each cell toggles. A sufficient toggling is further ensured by $n$-detect test.

Clock nodes are selected by applying test patterns into the netlist on balancing mean dynamic power across nodes. The circuit netlist and initial transition patterns are the candidates to select several nodes in the clock tree. If the mean dynamic power balancing does not ensure sufficient equality of blocks, shifting the flip-flop (FF) among adjacent blocks provides partitioned blocks' adequateness. Gating hardware, interconnections, and reconnection of clock signals are added to the selected blocks. Clock signal reconnections require if FFs are shifted to balance blocks in power equality. Such shifting is realized by Engineering Changed Order (ECO) to have minimal impacts on the design.

In the design phase, analyzing the maximum number of partitions for a given netlist is performed. Based on the analysis, the clock gating controller circuit can be put into the design to gate several clock buffers in the clock tree. It synthesizes the netlist with the clock gating controller and determines the clock buffer position after the placement. After the clock tree is synthesized, some clock buffers can be modified to the clock gating controller option. It may change the connection of FFs to the clock tree by taking care of the position of FFs. Components can be added between the clock gating controller and the design. An ECO option is utilized to re-route the modified design to exhibit a minimum impact on clock signal routing.

### 3.3 Clock tree based circuit partitioning

**3.3.1 Clock-tree partitioning.** The partition algorithm is from the previous work [34]. A sample clock-tree is shown in Fig 4 to understand the partitioning algorithm. Assuming the clock tree has 23 clock nodes or buffers where $R1$ is the root buffer. The rectangular boxes indicate flip-flops. There are twelve leaf nodes of clock buffers where the FFs are connected directly.

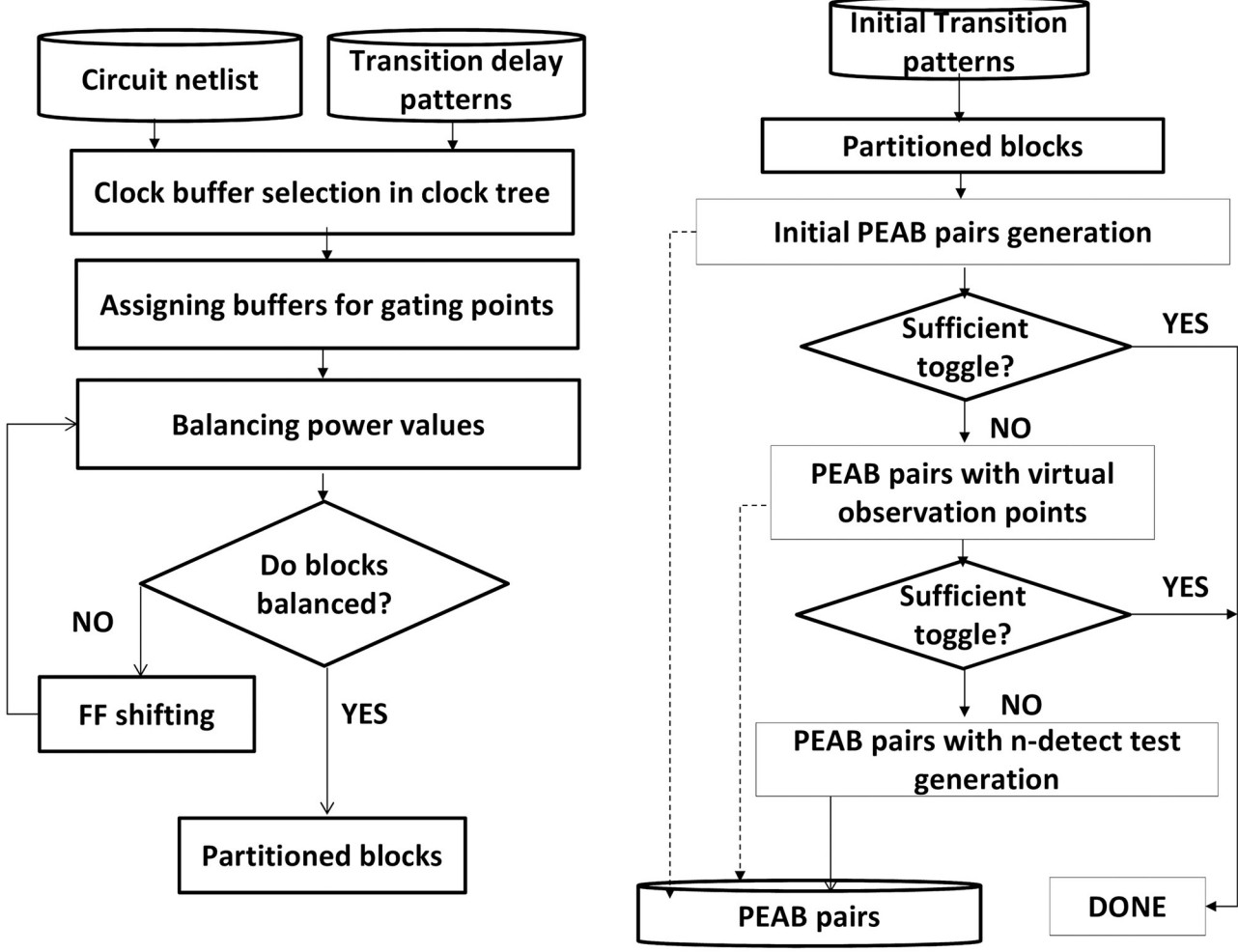

**Fig 3. Flowchart representation of the proposed PEAB pair generation.**

The clock-tree partitioning algorithm selects gating points in the clock-tree in a top-down tree traversal fashion. The clock tree with the number of clock nodes or buffers and the number of blocks are given as inputs. The function recursively determines gating points from the root of the clock tree. If the number of leaf nodes is less than a given number of blocks, the process assigns gating points to all the leaf nodes. If there are more leaf nodes, the number of children nodes are assigned to gating points based on their mean dynamic power values. Since the transient power when applying a pattern consists of dynamic power from an activated block and leakage power from an entire circuit, partitioning affects only dynamic powers.

If the number of required blocks is greater or equal to the number of clock nodes, the algorithm distributes gating points to the children according to their dynamic power values. Alternatively, if two or more gating points are assigned to one child, the algorithm recursively assigns gating points to a sub-clock tree rooted by the child. If the number of required blocks is less than the number of clock nodes, the algorithm selects children nodes using a bin-packing algorithm. Fig 4 displays that seven clock gating points are selected from the 23 clock nodes. Nodes N12 and N13 combine to a single node by a bin-packing algorithm. The red-colored circular nodes are selected as gating points in the clock tree.

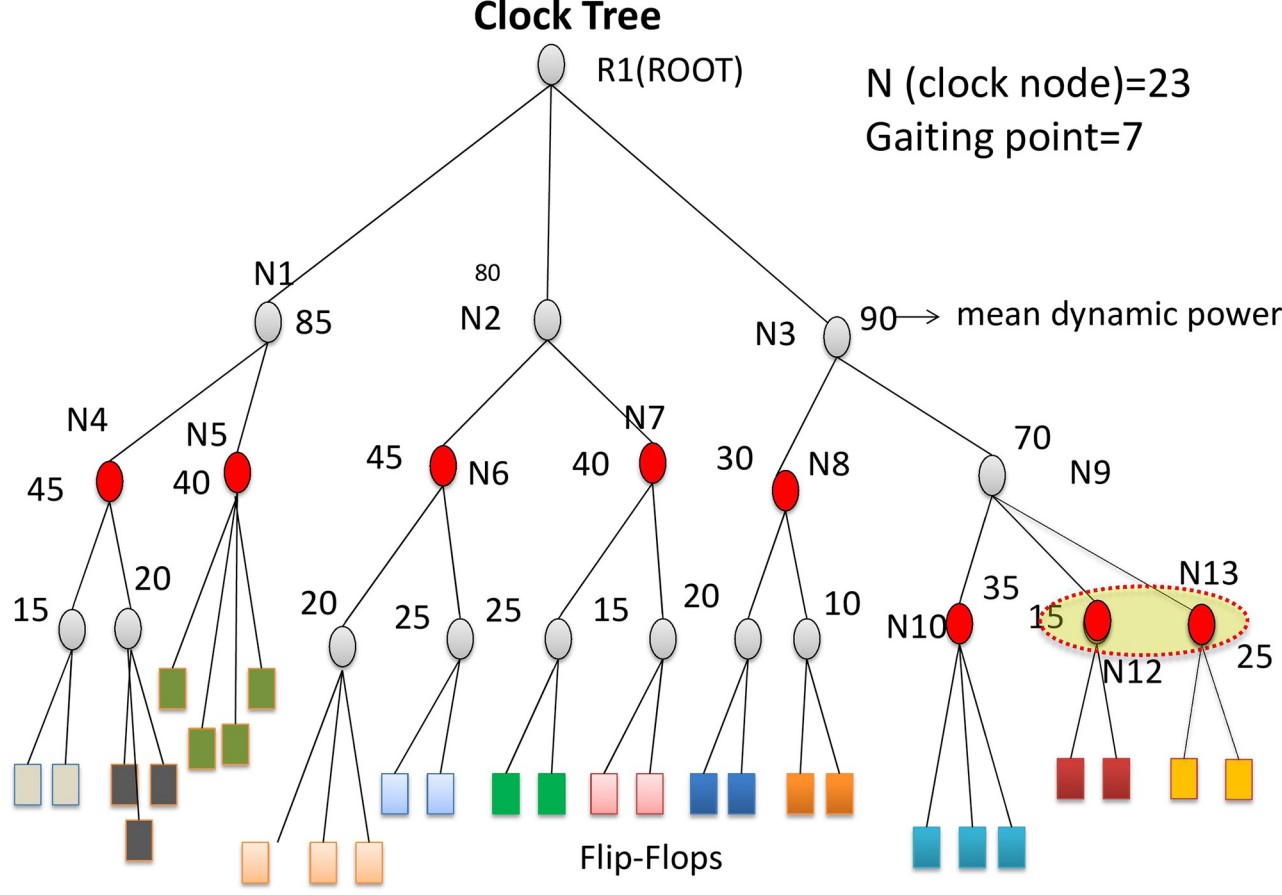

**Fig 4. The clock-tree partitioning algorithm that partitions clock nodes into blocks.**

**3.3.2 Block selection.** As mentioned above, the proposed PEAB pair generation blocks are selected through clock nodes in the clock tree from a given number of blocks in a chip. Transition delay fault patterns are applied to scan chains through a clock gating controller in such a fashion that a single clock node can get the clock signal where the remaining nodes under the same parent remain inactive. This pattern application activates several FFs in the chip; thus, it activates a group of combinational cells. A set of blocks is selected so that it can cover all FFs in a chip. The dynamic power consumption is considered for selecting blocks as the leakage power is similar for any node in the chip or independent of individual block power. Assuming *n* blocks are required for a given chip. If there are *m* clock nodes in the clock tree of the chip, a tree traversal algorithm is applied to select *n* nodes out of *m* clock nodes. The selected *n* nodes are the initial blocks measured by each block's power consumption, while others remain inactive. If the initial blocks deliver much deviated mean power among blocks, FF shifting across adjacent blocks is performed to produce more balanced blocks.

**3.3.3 Shifting FF across adjacent blocks.** FF shifting is an iterative shifting of FFs happens by calculating the standard deviation until it accomplishes a preset least deviation factor to outline the mean power gap across blocks. The deviation factor is updated every time a FF shifts to the adjacent block. As the deviation factor may change with circuit attributes, a further shifting may keep the deviation factor constant. Such cases are terminated with a user-defined limit to obtain balanced blocks. The most deviated block is adjusted considering two cases: a

FF with low mean power shifts out the nearest FF to the adjacent block for the maximum mean power; for the minimum mean power value of a block, the nearest FF with high mean power from the adjacent block is shifted in. Each iteration shifts one FF utilizing the geometric position of the FF in the layout. The shifting means reconnection of shifted FF to the new clock buffer. The impact of shifting in the circuit layout is minimized through ECO.

### 3.4 Test pattern selection

**3.4.1 Test generation.** The selected test pattern set ensures a minimum of a single transition of each cell. If any cell remains untoggled, a virtual test point insertion is proposed so that the minimum requirement of cell toggling can be met. The toggling sufficiency of PEAB pairs is further enhanced by generating $n$-detection tests repeatedly. The virtual observation points are inserted at the gate output, which does not toggle with ATPG patterns. The main role of inserting VOPs is to toggle cells that are not toggled with the ATPG patterns. Some faults may be undetectable, which means no patterns exist that can detect that particular fault. An undetectable fault is defined to be the case of no test pattern being able to simultaneously activate a fault and create a sensitized path to a primary output. VOPs are inserted at the output of untoggled gates and treated as fault sites. These VOPs force the ATPG to propagate transitions to outputs and increase the number of transitions or toggles. The VOP is temporarily inserted into the netlist for the sole purpose of generating test patterns by the ATPG, thus not requiring any extra hardware like test point insertion does, as can be seen in Fig 5.

An additional test pattern is selected if any additional cell is toggled at every iteration. Every time a new cell gets toggling is adopted as a new test pattern and included in the PEAB pairs set. In each repeated process, the number of transitions in the initial PEAB pairs set is checked. An additional test pattern generation step with a higher value of desired toggles is introduced if it gets insufficient toggling counts. The iteration process terminates after getting sufficient toggling or reaching some iteration limit.

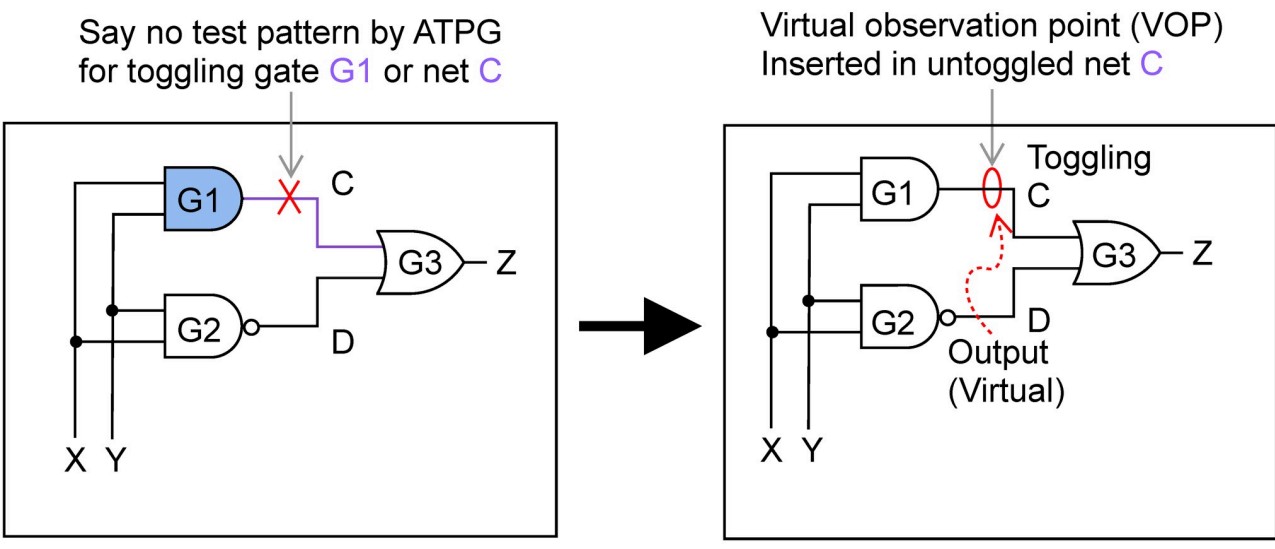

(a) ATPG can not generate transition pattern to sensitize line C.

(b) VOP insertion in the temporary netlist for test generation only.

**Fig 5. VOPs insertion in untoggled cells or nets: (a) ATPG patterns can not sensitize net C, (b) VOP insertion can ensure the untoggled net C toggling.**

**3.4.2 Sufficient toggling.**   The selected PEAB pairs impose a minimum of 1-transition for each cell by inserting VOPs and *n*-detect test. The role of inserting VOPs is to toggle cells that are not toggled through the ATPG patterns. In general, an undetectable fault in a gate output corresponds not to toggle that gate. The undetectable fault is defined if no test set can simultaneously activate a fault and create a sensitized path to a primary output. To get a full test coverage, it is common to insert test points into circuits, which results in some hardware overhead. Test point insertion is a design constraint, and the overhead increases linearly for large circuits.

On the other hand, the VOP is a net connection between the output of an untoggled gate termed as fault sites and the output port of a circuit netlist. Through the VOP, it is possible to ensure toggle of the untoggled cell even if it can increase the number of toggles since the insertion cuts the amount of effort the ATPG involves to propagate a transition to the output port. No extra hardware cost is required as VOPs are temporarily inserted into the netlist for test generation purposes only. In the proposed technique, observation points are inserted into the netlist virtually to generate patterns. Suppose an output net *i* of a gate *G* in the original netlist cannot be toggled with ATPG patterns since no test exists for that gate. If it inserts an observation point temporarily in *i* by connecting *i* with the output port, the ATPG can generate test cases to toggling *G*. Such generated test patterns can apply to the original netlist without VOPs to ensure the toggling of that net *i* without any hardware cost like the test point insertion method.

## 3.5 Power equal blocks

Several power simulations are run with transition patterns to identify each equal power pair of patterns that corresponds to adjacent blocks. Its toggling sufficiency checks every iteration of generating PEAB pairs. If it can consider any block in the chip, the PEAB pair count increases compared to adjacent consideration. This limitation comes from the constraint of neighborliness consideration. The less count of pairs may reduce the toggling coverage for the entire pair set. This toggling insufficiency is compensated by running the *n*-detect test into the VOP inserted netlist. In each run, the *n* value is increased by one to get sufficient toggling. Only a subset of *n*-detect tests are used to select PEAB pairs.

## 3.6 Current sensors: Parametric measurement

The proposed detection technique is an on-chip measurement of the peak current of a block by placing 16 current sensors in the empty spaces so that they are near the power grid and can measure the sinking current from each block. In [61], multiple current sensors are placed in different empty spaces of segments near the power rail. We consider the similar placement but in a distributed way so that multiple adjacent blocks can share a single current sensor, and the systematic variation effect can be minimized while comparing the PEAB pairs. The layout of the AES chip with the proposed Built-in-Current Sensor (BICS) is shown in Fig 6. Sixteen current sensors are placed into the power rail's empty spaces, thus delivering minimum hardware overhead. Red circular symbols represent the current sensors. As the AES chip is partitioned into 42 blocks, the clock gating controller has access to all blocks. The test controller can control all 42 gating nodes and 16 current sensors.

The on-chip detection circuit consists of on-chip sensors, memory to store data, and power comparison hardware [62]. A sensor consists of an analog-to-digital (ADC) converter that converts the current sensor's analog current value to digital bits. The ADC's output power value is stored in a 22-bit memory unit for comparison with the power value of the target block. The ADC unit consists of 16 D-FFs and an operational amplifier. We designed an 8-bit

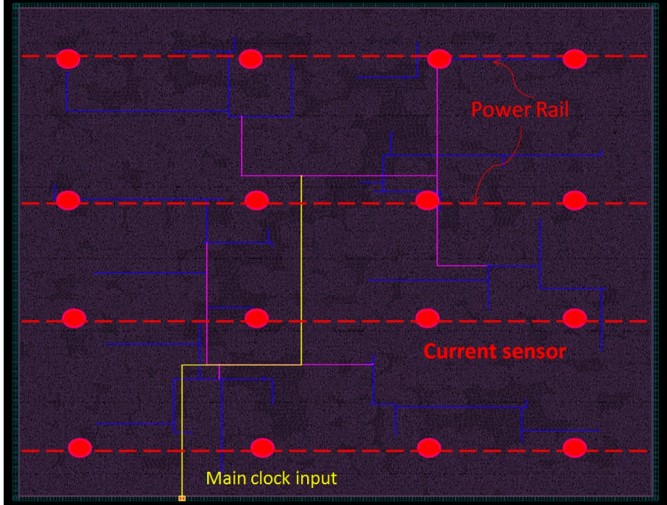

**Fig 6. The layout of the AES chip with 16 current sensors.**

comparator circuit comprised of 154 transistors. Fig 7 shows the hardware requirement for the proposed DMST approach. There are 16 current sensors embedded in the AES chip; each of them consists of 6 transistors. When a PEAB pair (tuples $(B_i, t_i)$ and $(B_j, t_j)$) is applied to the adjacent blocks $i$ and $j$, first, the dynamic switching current of block $i$ is measured by the

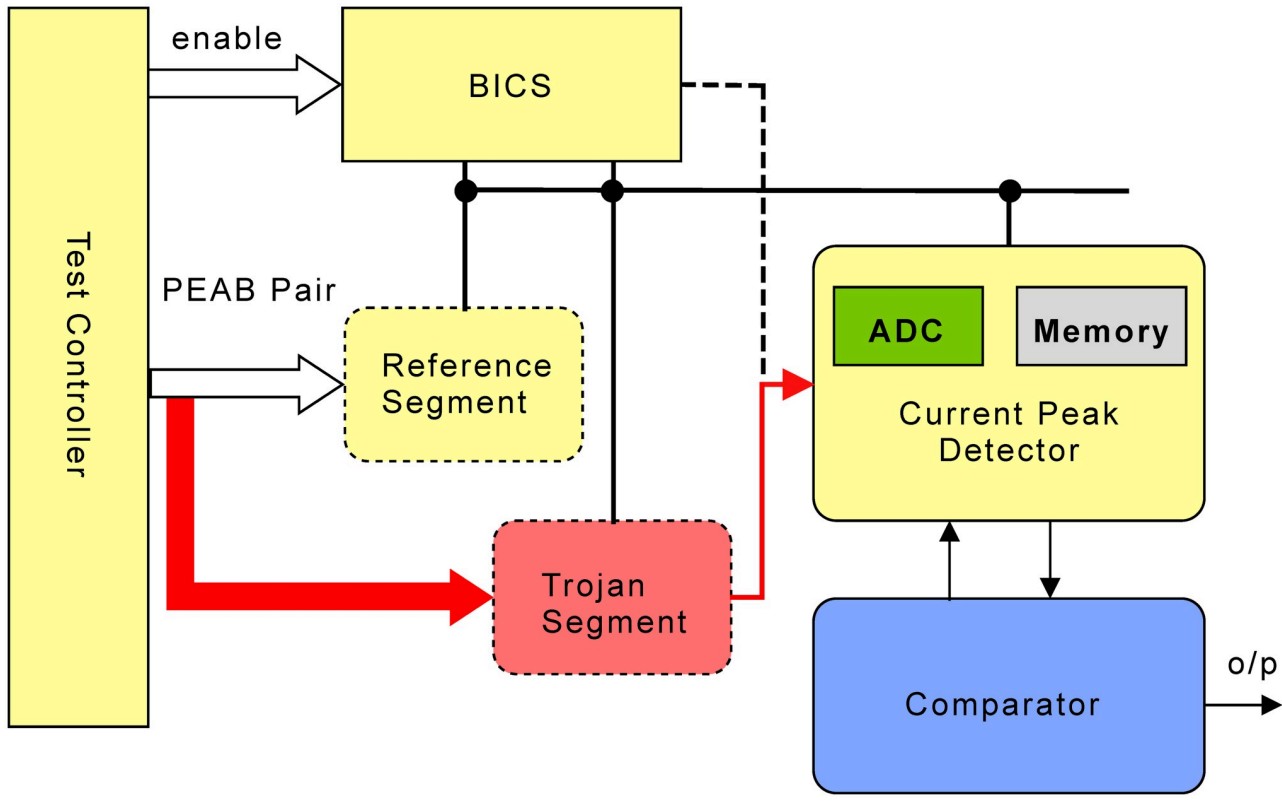

**Fig 7. Self-referencing with built-in current sensors and memory hardware.**

current sensor and converted to the power value and stores in the memory. Second, it measures the power value of the block *j* and stores it in the memory. The compactor circuit takes power values from the memory unit and calculates the CPR values. The measured power value is considered seven digits after decimal; therefore, we need 42 bits with two slots. At every operation, the data in the memory is replaced with a new one; thus, the memory size is reduced.

## 3.7 Design of test moods

The proposed method applies a test pattern through a given number of scan chains. The clock gating controller unit activates a target block while keeping other frozen. Each test pattern in a pair contains a block ID that initiates a one-hot operation. The proposed two test moods by selecting a flag: the flag's actual value ensures the Trojan detection mood where the alternative mood delivers fault detection mood. The Trojan detection mood is utilized only a single time after manufacturing the chip and deactivated forever. The fault detection mood is working for online detection during the chip is in operation. As the Trojan is inserted in the manufacturing step and unable to be manipulated after fabrication, the Trojan detection test mood is only once. Any manufacturing defects such as aging and other parametric defects can happen during the chip operation; thus, it considers the parametric fault detection mood online.

## 3.8 Process variation window of power equal blocks

A mathematical analysis shows how the power equal blocks in self-referencing deliver a threshold constraint experiencing inter-chip and intra-chip variations. In this proposed DMST method, the intra-chip systematic variation effect is diminished by considering the spatial correlation between two adjacent blocks in a chip. The random variation effect is addressed by adhering to sufficiently large blocks. A cross-referencing of a pair of blocks in a chip prevents the impact of the inter-chip variation effects. A mathematical analysis can approximate Intra-chip variation effects.

The following notations are used to derive the process variation threshold from identifying defects and Trojans.

- $W_m(B)$: Total measured power (dynamic and leakage power) when a block $B$ is activated.

- $W_{nom}(B)$: Nominal power when a block $B$ is activated.

- $W_{nom\_dy}(B)$: Nominal dynamic power when a block $B$ is activated.

- $W_{nom\_leak}(C)$: Nominal leakage power for a chip $C$.

- $\nabla_{inter}$: a random variable to represent a factor of deviation of total power from a nominal power instigated by the inter-chip variation.

- $\theta_{inter}$: a random variable to represent a factor of deviation of leakage power from a nominal power caused by an inter-chip systematic variation.

- $\nabla_{i,sys}$: a random variable to represent a factor of deviation of total power from a nominal power caused by an intra-chip systematic variation for a block $B_i$.

- $\nabla_{ij,sys}$: $\nabla_{i,sys} - \nabla_{j,sys}$ for two blocks $B_i$ and $B_j$.

- $\nabla_{i,ran}$: a random variable to represent a factor of deviation of total power from a nominal power caused by an intra-chip random variation for a block $B_i$.

- $\forall_{ij,sys}$: a value to represent an acceptable range of $\nabla_{ij,sys}$, i.e., $|\nabla_{ij,sys}| \leq \forall_{ij,sys}$. For example, $\forall_{ij,sys} = 3\sigma_{ij,sys}$ can be considered where $\sigma_{ij,sys}$ is a standard deviation of $\nabla_{ij,sys}$.

- $\forall_{ran}$: a value to represent an acceptable range of $\nabla_{i,ran}$ for any $i$, i.e., $|\nabla_{i,ran}| \leq \forall_{ran}$.

- $\varphi$: measurement error margin for for current sensors.

- $B_i^F$: a faulty block.

Transistor size and the process technology determine the amount of effects of PV on the leakage and the dynamic power of transistors with a variance. Assuming $W_m$ is the measured power when its nominal power $W_{nom}$ is influenced by a factor $\forall$, therefore, $|W_m - W_{nom}| \leqslant \forall * W_{nom}$. It implies that the maximum power difference in measured chips can be $2\forall$. Therefore, in the DMST, the nominal power difference of two blocks measures small deviations due to variations; conversely, it measures sufficient difference for the defects. To substantiate this relationship, the proposed PEAB pair based DMST approach utilizes the comparative power ratio (CPR) defined by (1), where $t_i$ and $t_j$ are the applied test patterns in blocks $i$ and $j$.

$$CPR(t_i, t_j) = \frac{W_{nom}(B_i^F) - W_{nom}(B_j)}{W_{nom}(B_j)} \qquad (1)$$

$t_i$ and $t_j$ can be identical or two different test patterns for two different blocks in a chip. As the measured power $W_m(B_i)$ differs from its nominal $W_{nom}$ power value with factors of inter-chip $\nabla_{inter}$ and intra-chip $\nabla_{i,intra}$ variations, the total measured power for $B_i$ can be written as (2).

$$W_m(B_i) = (1 + \nabla_{inter} + \nabla_{i,intra}) W_{nom}(B_i) \qquad (2)$$

Similarly, the measured power of $R_j$ is as (3).

$$W_m(B_j) = (1 + \nabla_{inter} + \nabla_{j,intra}) W_{nom}(B_j) \qquad (3)$$

If two adjacent blocks are compared in a PEAB pair ($W_{nom}(B_i) = W_{nom}(B_j)$), an intra-chip based correlation exists in between the power consumption of two adjacent blocks as can be written as (4). The correlation impact is dominated by dynamic power not by leakage as it is consumed in entire chip.

$$W_m(B_i) - W_m(B_j) = (\nabla_{i,intra} - \nabla_{j,intra}) W_{nom}(B_j) \qquad (4)$$

A power mismatch due to fault or Trojans in $B_i$ can indeed be identified if $W_m(B_i^F) - W_m(B_j)$ for a PEAB pair in the faulty chip $C^F$ exceeds a minimum threshold value. Intra-chip variations determine this threshold value. Therefore, the threshold condition can be written as (5) to detect faults.

$$W_m(B_i^F) - W_m(W_j) > (2\forall_{intra} + \varphi) W_{nom}(B_j) \qquad (5)$$

Here, $\forall_{intra}$ is the intra-chip variation. Again, $W_m(B_i^F)$ differs by a ratio of $\nabla_i (= \nabla_{inter} + \nabla_{i,intra})$ ($\nabla_{inter}$: inter-chip variation factor, $\nabla_{i,intra}$: intra-chip variation factor for a faulty block) from its nominal $W_{nom}(B_i^F)$ by (3) and $W_m(B_j)$ differs by a factor of $\nabla_j (= \nabla_{inter} + \nabla_{j,intra})$ from a nominal power $W_{nom}(B_j)$ by (4), where $\nabla_{j,intra}$ denotes the intra-chip variation component for the fault

free region.

$$W_m(B_i^T) = (1 + \nabla_{inter} + \nabla_{i,intra})W_{nom}(B_i^F) +$$
$$(1 + \theta_{inter})W_{nom\_leak}(C^F) \tag{6}$$

$$W_m(W_i) = (1 + \nabla_{inter} + \nabla_{j,intra})W_{nom}(B_j) +$$
$$(1 + \theta_{inter})W_{nom\_leak}(C) \tag{7}$$

Assuming the leakage power contribution by the Trojan is negligible, therefore, $W_{nom\_leak}(C^F)$ is equal to $W_{nom\_leak}(C)$. Incorporating (3) and (4), (2) can be represented as *CPR* to satisfy following conditions in (8, 9, 10, and 11) to get the final condition in (5).

$$(1 + \nabla_{inter} + \nabla_{i,intra})W_{nom}(B_i^F) - (1 + \nabla_{inter} +$$
$$\nabla_{j,intra})W_{nom}(B_j) > (2\forall_{intra} + \varphi)W_{nom}(B_j) \tag{8}$$

$$(1 + \nabla_{inter} + \nabla_{i,intra})W_{nom}(B_i^F) > (1 + \nabla_{inter} +$$
$$\nabla_{j,intra} + 2\forall_{intra} + \varphi)W_{nom}(B_j) \tag{9}$$

$$(1 + \nabla_{inter} + \nabla_{i,intra})W_{nom}(B_i^F) - (1 + \nabla_{inter} +$$
$$\nabla_{i,intra})W_{nom}(B_j) > (1 + \nabla_{inter} +$$
$$\nabla_{j,intra} + 2\forall_{intra} + \varphi)W_{nom}(B_j) -$$
$$(1 + \nabla_{inter} + \nabla_{i,intra})W_{nom}(B_j) \tag{10}$$

$$W_{nom}(B_i^F) - W_{nom}(B_j) >$$
$$\frac{(\nabla_{j,intra} - \nabla_{i,intra} + 2\forall_{intra} + \varphi)W_{nom}(B_j)}{(1 + \nabla_{inter} + \nabla_{i,intra})} \tag{11}$$

$$CPR(t_i, t_j)\left(= \frac{W_{nom}(B_i^F) - W_{nom}(B_j)}{W_{nom}(B_j)}\right) >$$
$$\frac{\nabla_{j,intra} - \nabla_{i,intra} + 2\forall_{intra} + \varphi}{(1 + \nabla_{inter} + \nabla_{i,intra})} \tag{12}$$

Assuming the spatial correlation factor $\nabla_{sys}$ of the intra-chip variation is a difference of two systematic components ($\nabla_{j,intra\_sys} - \nabla_{i,intra\_sys}$). Assuming $\forall_{ij,intra\_sys}$ denotes the realistic worst-case value of $\nabla_{ij,intra\_sys}$ with a standard deviation of $3\sigma_{ij,intra\_sys}$. $\sigma_{ij,intra\_sys}$ is the standard deviation of the distribution of $\nabla_{ij,intra\_sys}$. In another case, the effects of random deviations of $\nabla_{i,intra\_ran}$ and $\nabla_{j,intra\_ran}$ are independent of correlation. Therefore, the worst-case difference is $2\forall_{intra\_ran}$ under the random intra-chip variation.

The detection threshold in (2) and the *CPR* in (5) can be rewritten as in (13) and (14) respectively by putting $(\nabla_{j,intra} - \nabla_{i,intra}) = \nabla_{ji,intra}$.

$$W_m(B_i^F) - W_m(B_j) >$$

$$(\forall_{ij,intra\_sys} + 2\forall_{intra\_ran} + \varphi)W_{nom}(B_j) \tag{13}$$

$$W_{nom}(B_i^F) - W_{nom}(B_j) >$$

$$\frac{(\nabla_{ji,intra} + \forall_{ij,intra\_sys} + 2\forall_{intra\_ran} + \varphi)W_{nom}(B_j)}{(1 + \nabla_{inter} + \nabla_{i,intra})} \tag{14}$$

Eq (13) dictates that if the nominal dynamic power block is minimal with a more significant measured power difference, which can identify Trojans significantly with the PEAB pair. If the Trojan is into two neighboring regions in an equal number of similar gates and two test patterns in a PEAB pair simultaneously activate them, the PEAB pair cannot detect Trojans.

The notation of $(\nabla_{j,intra} - \nabla_{i,intra})$ in (14) consists of two terms: the systematic $(\nabla_{sys})$ and the random $(\nabla_{j,intra\_ran} - \nabla_{i,intra\_ran} = \nabla_{ran})$ effects of the intra-chip variation. Therefore, (14) can be rewritten as (15) by incorporating the definition of *CPR* in (1).

$$CPR(t_i, t_j) >$$

$$\frac{\nabla_{sys} + \nabla_{ran} + \forall_{ij,intra\_sys} + 2\forall_{ran} + \varphi}{(1 + \nabla_{inter} + \nabla_{i,intra})} \tag{15}$$

The proposed neighboring region comparison minimizes the systematic variation effect. Therefore, the intra-chip systematic variation factors $\forall_{ij,intra\_sys}$ and $\sigma_{ij,intra\_sys}$ can be approximated sufficiently small. Depending on the technology scale, the intra-chip systematic and random variations deliver different values. The random part predominates as technology scales while being almost equal at the transistor level for 65 nm technology [63]. The random variation effect can be addressed by activating multiple cells together [64]. For a given technology, if $h$% of intra-chip variation is for random variation and the intra-chip standard deviation is $\sigma_{i,intra}$, then the intra-chip random standard deviation $\sigma_{i,intra\_ran}$ for $B_i$ can be written as in (16).

$$\sigma_{i,intra\_ran} = \frac{h * \sigma_{i,intra}}{\sqrt{n}} \tag{16}$$

## 4 Results

For the validation of the proposed method, faults of various types are inserted. Two types of Trojans termed TR1 and TR2, considering with modification from Trust-HUB benchmarks [60] are inserted as Trojans. TR1 and TR2 are combinational type sequential and sequential type of combinational Trojans respectively [65]. Two more sequential type Trojans without any modification from Trust-Hub named AES-T300 and AES-T400 are inserted. On the one hand, the fault injections are performed to realize stuck-at-0, stuck-at-1, propagation delay, bridging faults, and others that deliver different power consumption of the faulty blocks while comparing to reference blocks. As any of the mentioned faults may cause changing the delay profile of the signal propagating from input to the output paths, and the paths' consumed power may differ from the references. The original designs are synthesized using the Synopsys design compiler and IC compiler with a 90nm technology library.

### 4.1 Experiments for design

The assumption was to attain a Gaussian distribution of power values for the ATPG patterns. It observed that the distribution of power values for the ATPG test patterns holds the assumption. In the proposed method, it adjusts blocks to obtain more PEAB pairs to achieve more similar mean powers. If the total power values for each block have a Gaussian-like distribution, many test patterns consume similar power to their mean value. The experimental evaluation displays that the initial 50 blocks of the AES-128 circuit follow such distribution. The initial 50 blocks deliver some deviated blocks in terms of mean dynamic power. The deviation is addressed by combining 50 blocks to 42 blocks and further addressed by shifting 100 FFs to the adjacent blocks. Fig 8 displays the first 10 out of 42 selected blocks showing Gaussian distribution. The geometric position of flip-flops in the layout of the AES-128 circuit is within a 1.1 mm × 1.53 mm rectangle. Two blocks are identified as to be adjacent if the maximum distance between flip-flops in the two blocks is 0.068mm or less. With the initial test patterns from ATPG, the AES-128 fails to toggle 131 cells. After inserting 131 VOPs into the netlist, the generated test patterns can toggle all cells.

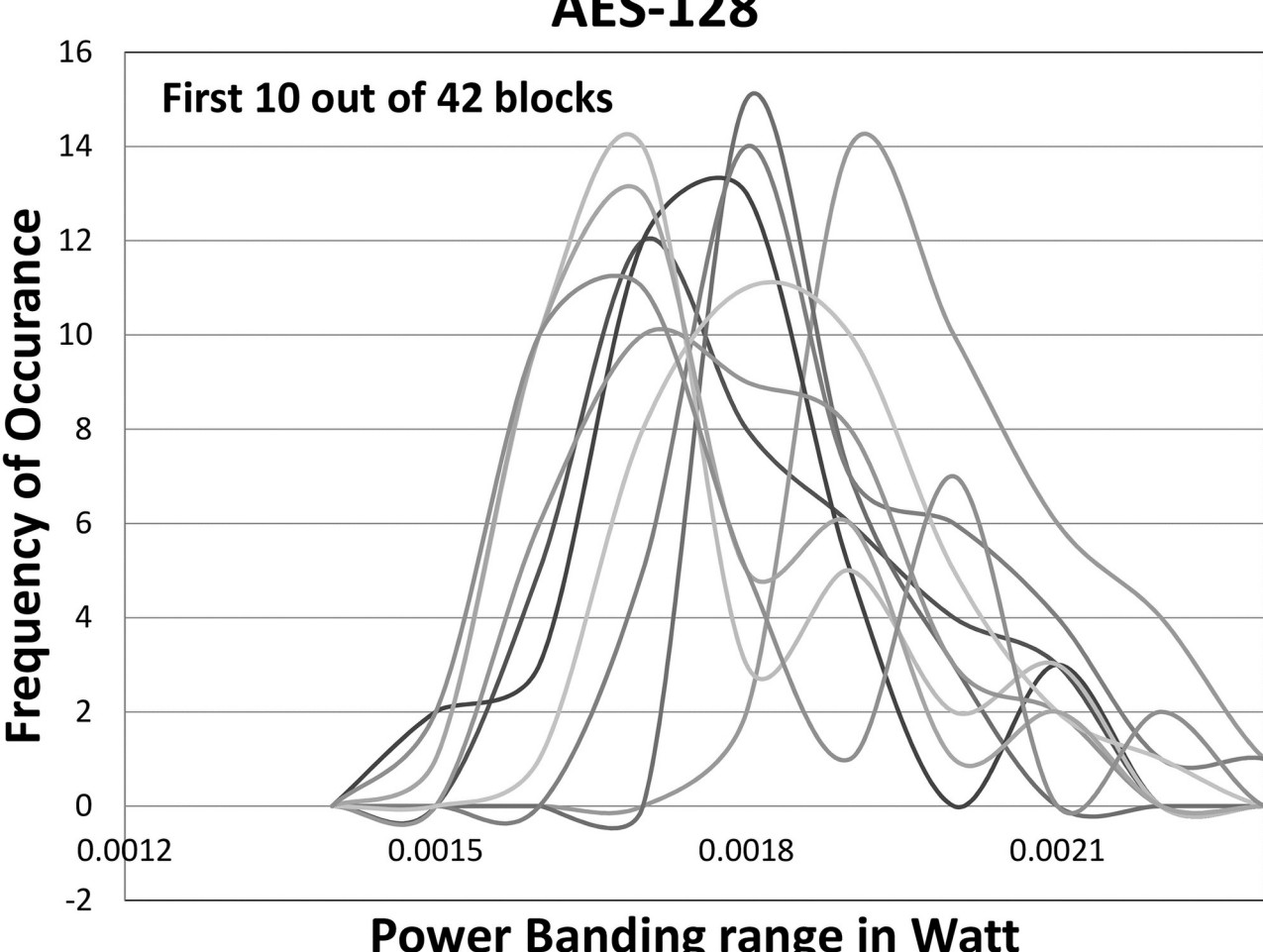

**Fig 8. Distribution of nominal power values of first 10 blocks for the AES-128 show a Gaussian distribution.**

**Table 1. Area, power and delay overhead of the AES-128 circuit with clock skew information.**

| Attributes | Value |
|---|---|
| Initial end node clock buffers or blocks | 50 |
| Selected blocks considering mean dynamic power | 42 |
| Number of shifted flip-flops | 100 |
| Hardware overhead | 0.134% |
| VOP insertion overhead | 0.0% |
| Average power per block (overhead) | 0.542% |
| Worst case delay overhead | 0.0124% |
| Longest path delay without gating | 0.592ns |
| Longest path delay with gating controller | 0.592ns |
| Shortest path delay without gating | 0.525ns |
| Shortest path delay with gating controller | 0.523ns |

The proposed technique distributes clock gating hardware with shifting of flip-flops to get more PEAB pairs. Table 1 shows an initial number of clock buffers in the clock tree, blocks after gating point selection, shifting of flip-flops with clock skew, and test controller hardware overhead. We evaluate the hardware overhead for the proposed DMST design. Hardware overhead of 16 current sensors, clock gatting controller, comparator, and memory units are also presented. We do not consider the overhead of scan design. Our assumption is that the scan chain technology is a well-established testing method, and researchers in this field consider the extra hardware cost due to design-for-testability purposes. Therefore, the scan chain overhead is not considered; we believe it is already included in the design. However, the clock gating controller, memory, embedded current sensors are the extra hardware to obtain the power side-channel parameters. Hardware overhead of o.134% includes test controller (current sensors and clock gating), memory unit, and comparator circuit. VOP insertion does not claim any hardware overhead as all the test points are inserted virtually for test pattern generation. The average power per block is without the scan chain power consumption. Clock skew information ensures the setup and holds time violation indicator, which is met in the experiment. The critical path delay is unchanged for the DMST design, and the slack is met at 4.55 ns.

A detailed result is displayed in Table 2. With the scan chain design, the hardware overhead is 5.239%, while it is only 0.134% without scan design. We assume the scan design is already considered for the manufacturing test. The test controller (clock gating and current sensors) delivers more overhead (0.091%) compared to memory, ADC, and comparator circuits. The test controller overhead is approximately 97%, which means the overall hardware penalty is for the proposed DMST architecture controller unit. The test controller applies the PEAB pair to the current sensors through the scan chain, and the response is stored in the memory unit.

**Table 2. A detailed result of hardware overhead, power and delay with and without scan design.**

| Event | Complete design | Scan design | Test contr. | Memory | ADC | Comparator circuit |
|---|---|---|---|---|---|---|
| Total Area in um width | 2163275.57 | 110275.2 | 1968.58 | 520 | 241.34 | 163.18 |
| Area Overhead | 5.239% | 5.097% | 0.091% | 0.024% | 0.011% | 0.007% |
| Avg. power per block in mW | 17.521 | 3.627 | 0.0647 | 0.0171 | 0.0079 | 0.0053 |
| Power overhead per block | 21.073% | 20.703% | 0.369% | 0.098% | 0.045% | 0.031% |
| Critical path delay in ps | 592 | 178.38 | 50.078 | 13.228 | 6.139 | 4.151 |
| Delay overhead | 42.56% | 30.132% | 8.459% | 2.234% | 1.037% | 0.701% |

The area overhead of memory and comparator is a sufficiently little value compared to scan design. The observation on hardware overhead delivers a small percentage of overhead and can ignore for the AES or a larger circuit. The average power per block for the complete design is 7.521mW, in which a large portion (3.627mW) is for the scan chain. The power overhead is 0.543% without scan design. Similarly, the delay of the whole circuit is 592ps, and the overhead is 12.42% in the pico-second range.

In our experiment, we insert the maximum inter-chip variation of 30% ($3\sigma$) and the maximum intra-chip variation of 20% ($3\sigma$) to demonstrate the effectiveness of the proposed method under enormous variations experienced in the real chip. The *CPR* variation threshold is evaluated with certain inter-chip and intra-chip variation factors to identify the number of faults or Trojans. It considers the standard deviations $\sigma_{sys}$ and $\sigma_{ran}$ have 1:1 ratio of the standard deviation $\sigma_{intra}$, thus it can be estimated roughly $\sigma_{sys} = \sigma_{ran} = 0.7^*\sigma_{intra}$. The PEAB pairs set can toggle 1,238 cells of the AES-128 per block with an intra chip random standard deviation of $\sigma_{i,intra\_ran}$ for the average variation case from (15). Considering the current technology trend, the standard deviation of spatial correlation $\sigma_{ij,intra\_sys}$ can be assumed to be 5% of $\nabla_{ij,intra\_sys}$. The identification of faulty chips for four variation cases is evaluated. The cases are 20% inter with 10% intra (low), 20% inter with 15% intra (medium), 20% inter with 20% intra (high) and 30% inter with 20% intra (very high). The Monte Carlo simulation of 60,000 samples ensures the optimum identification levels for particular threshold values of *CPR* in Fig 9 with a measurement error $\varphi$ of 5%. The graph shows that 99.9% identification is achieved with an *CPR* of 7.3% for the low, 8% for the average or medium, 9.2% for the high and 10% for the very high variation cases.

## 4.2 Parametric test fault injection

Parametric fault identification is tested through several faults injection into the AES-128 optimized netlist in block-4 and 11. U47581 is two inputs NOR gate and gets toggling for several patterns in block-4. It is forcefully short-circuited the output of this gate into the Vdd line to achieve a stuck-at-1 fault. This fault initiates few extra gates toggling in the signal propagation path. The extra switching contributes more dynamic power in block 4 for several patterns. The maximum comparative power ratio of 8.5% is achieved in PEAB pair-8. Similarly, other faults in the netlist can be inserted to measure the CPR, which are listed in Table 3.

## 4.3 Experiments with Trojans

Two types of Trojans: TR1 and TR2, are inserted in the AES-128 circuit. TR1 is the two 8-bit comparators with two flip-flops, and TR2 is a 16-bit comparator with sequential 4-bit counter

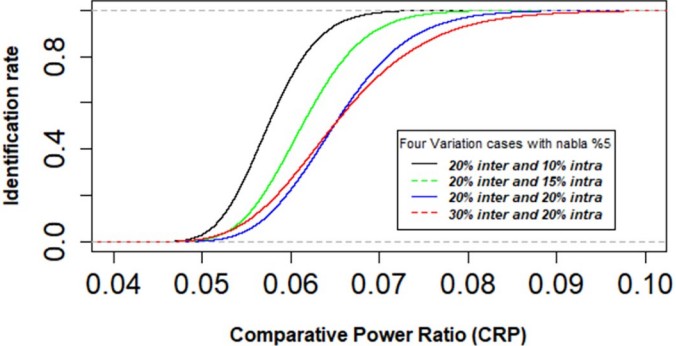

**Fig 9. Cumulative distribution function of 60,000 sample for identification of faulty chips.**

**Table 3. Faulty AES chip identification for different faults and Trojans under low, medium, high and very high variation cases.**

| Events | Max CPR | Identification in different variation cases | | | |
|---|---|---|---|---|---|
| | | Low | Medium | High | Very high |
| Stuck-at-fault | 8.50% | 100% | 100% | 99% | 97% |
| Bridging fault | 9.10% | 100% | 100% | 100% | 98% |
| Propagation delay | 10% | 100% | 100% | 100% | 99% |
| Trojan TR1 | 10.30% | 100% | 100% | 100% | 99% |
| Trojan TR2 | 7.80% | 100% | 99% | 95% | 91% |
| Trojan AES-T300 | 13.57% | 100% | 100% | 100% | 100% |
| Trojan AES-T400 | 11.33% | 100% | 100% | 100% | 99% |

modified from the trust-Hub benchmark of AES-T100 and AES-T200 Trojan circuits. The detail of the modification is presented in our previous [66]. Another two types of Trojans (AES-T300 and AES-T400) are inserted without modification. The Trojan AES-T300 demonstrates an attack on the AES-128 block-cipher and its corresponding key schedule. The idea is to artificially introduce leaking intermediate states in the key schedule that depend on known input bits and key bits, but that naturally would not occur during regular processing of the cipher. The Trojan uses AND conjunctions to pairwise combine each key bit with another input bit. The output of the AND gates is then combined to the leaked intermediate value by XORing all of them. The Trojan leaks one byte of the AES round key for each round of the key schedule. The leakage circuit is a 16-bit shift register loaded with an initial alternating sequence of zeros and ones. The shift register is only enabled if the leakage circuit's input is one, which results in additional dynamic power consumption. AES-T400 is modulating with an unused pin on a chip generates an RF signal. This signal can be used to transmit the key bits. This attack is performed at 1560 kHz and can be received with an ordinary AM radio. The data carried by the AM signal needs to be easily interpreted by a human. A beep scheme is utilized where a single beep followed by a pause represents a '0' and a double beep followed by a pause represents a '1'. A description on detail implementation of AM transmission can be found at [67]. In this implementation, the Trojan gets activated when a predefined input plaintext is observed.

The detailed results are displayed in Table 3 considering four variation cases (low, medium, high, and very high). Results show the maximum CPR for inserted Trojans. For the TR1, AES-T300, and AES-T400 Trojans, the maximum CPR delivers almost 100% identification for all the variation cases. The sequential dominant Trojan TR2 gets less CPR value delivering 95% identification for high variation cases. In contrast, this identification rate down to 91% for the very high variation case means 54,600 out of 60,000 chips can be identified successfully. As it considers the sensor error rate 5%, the CPR for PEAB pairs cannot be less than 5%, which implies that a sensitive detection requires the design of highly precise sensors to reduce the CPR threshold further.

The analysis on various faults and Trojans indicates that an increased number of false-negative (faulty chips identified as a good one) occurrences may happen for similar faults or Trojan activation in adjacent blocks through a single PEAB pair. The AES-128 chip shows that in case of such faults or insertion of Trojans in two adjacent blocks of a PEAB pair, the probability of common activation is possible. The PEAB pair that can activate faults or Trojans equally to its pair of blocks cannot identify the mismatch. It randomly inserts the TR1 and AES-T300 Trojans in some nets of logic gates, which have chances to be activated commonly by several PEAB pairs. The evaluation provides interesting outcomes showing that some other PEAB pairs can deliver sufficient CPRs through the intended PEAB pairs fail to identify.

**Table 4. Comparison with the state of the art fault and Trojan detection of AES in dual mode self-test implementation.**

| Method name | Detection mode | Exploit mechanism | Design overhead |
|---|---|---|---|
| OBISA: Obfuscated built-in self authentication with wire-lifting [68] | Trojan Detection | Filler cell insertion and wire lifting | Delay = 4.08%<br>Power = 12.73%<br>Area = 0.42% |
| TPAD: Trojan prevention and detection [69] | Trojan detection using fault tolerant computing | functional and online detection technique | Delay = 0%<br>Power = 7-60%<br>Area = 7.4-165%<br>(based on design) |
| BISA: Built-In Self Authentication [70] | Trojan prevention | Filler cell insertion | Delay = 4.28%<br>Power = 14.38%<br>Area = 22.54% |
| K-security: 3D IC based circuit [71] | Trojan prevention | Wire lifting | Delay = 54%<br>Power = 114%<br>Area = 167% |
| Online error detection for AES core [72] | Fault detection | Parity code | Delay = 2.4%<br>Power = NA<br>Area = 16% |
| A-SOFT-AES: Self-Adaptive Software based AES [73] | Fault Detection | Soft implementation | Delay = NA<br>Power = NA<br>Area = 3% |
| Hybrid-AES: Pipeline based Hardware design [74] | Error Detection | Hybrid implementation | Delay = 1.27%<br>Power = NA<br>Area = NA |
| Proposed DMST: Dual mode self-test | Identification faults and Trojans in online | Variation aware self-referencing | Delay = 0.012%<br>Power = 0.543%<br>Area = 0.134% |

## 4.4 Discussion and comparison

Table 4 displays a comparison among different methods for identifying Trojans and faults in self-test for the AES circuit. As our proposed DMST is a novel method and has not been addressed yet in the literature, we make a comparison based on the performance, power, and area overhead of some self-test and hardware Trojan detection works. In [68], obfuscated built-in-self test is introduced to detect Trojans. They reduce the number of filler cell insertions so that the test structure's area, delay, and power are reduced significantly in the AES and DES chips. A significantly lower area overhead (0.42%) is reported in this work. However, the performance and power consumption are much higher. Our proposed DMST method delivers negligible area overhead (0.134%), which is more than three times smaller than that of [68]. The listed works [68, 70, 73] in Table 4 utilize the scan design for design-for-testability; however, they do not consider the scan overhead in the reported area, power, and delay overhead. The scan chain technology is a well-established testing method and is accepted as the design part for manufacturing tests.

We compare our work with Trojan detection approaches [69–71] in the AES chip. In [69], the Trojan detection technique (TPAD) is performed utilizing fault tolerating computing. TPAD is derived from the concept of concurrent error detection for fault-tolerant computing, thus delivering area overheads range from 114% to 165% with 34% to 61% power overheads. However, their re-configurable and split manufacturing techniques can reduce the area overhead to 7.60% with 0% delay. In [70], a built-In self-authentication method is presented to prevent inserting hardware Trojans. Their motivation is that the unused spaces in the circuit layout can be the best opportunity to insert Trojans. This method works by eliminating these spare spaces and filling them with functional filler cells instead of nonfunctional filler cells,

resulting in high overhead in the area (22.54%), power (14.38%), and delay (4.28%) for the AES chip. A Trojan prevention technique is introduced in [71]. Their wire lifting approach delivers huge overhead in area, power, and delay as multiple wires removing and reconnecting change the layout significantly.

Some significant fault detection in AES chip has been reported [70–74]. In [72], a self-test structure is introduced to identify faults in a 32-bit AES chip in the FPGA platform. The error detection scheme is used to detect faults. The reported area overhead of 163% and delay of 2.4% are larger values for a tiny 32-bit AES core. In [73], a software-based self-test AES method is presented. This technique is based on a pool of software-implemented fault-tolerance techniques that dynamically chooses the best one in terms of performance, cost, and fault-tolerance for a wide range of fault rates. Their reported area overhead is 3%. In [74], a hybrid of software and hardware design is proposed in the FPGA platform. They reduce the delay and area of the design-for-testability through a pipeline structure, analyzing and balancing the critical path and distributing the processing elements within each stage. The latency is 11 clock cycles in a 5.542 ns clock period that means the delay overhead is approximately 1.27%.

In our proposed DMST method, we catch both Trojans and manufacturing defects by a dual-mode self-test method. However, the test controller, memory unit, and comparator circuit deliver overhead on area, power, and delay reported in Table 4. Our hardware implementation consumes less power, delay, and low hardware overhead compared to other related works. We have inserted test points in the AES chip to get 100% test coverage with 0% area overhead. In state-of-the-art, none of the reported methods perform both the fault and Trojan test in the hardware domain through BIST. DMST is the first attempt to implement a stand-alone AES core in self-test to the best of our knowledge.

## 5 Conclusion

A parametric mismatch identification technique for the AES-128 chip is presented in this paper. Several technical designs are considered in power side-channel analysis to deliver a stand-alone and reliable AES core. A novel comparative power ratio-based identification threshold is designed to fit the simulated power to the actual chip outcomes. In the case of different types of faults and Trojans, this technique demonstrated 100% identification under the presence of worst process variation levels. Despite the fact even of considerable variation levels, the results showed a reasonable identification rate. The design overhead in area, power, and delay is negligible compared to other state-of-the-arts. If the measurement sensors' accuracy can be improved further, in that case, the identification threshold window can be reduced sufficiently, implying a small impact of fault in power, or a few Trojan gates can be identified under elevated process variations. The future scope of this DMST method to implement in the FPGA platform and identify the testing complexity. Some large-scale industrial chips may be tested with this DMST architecture to demonstrate the practicability of this technique.

## Acknowledgments

The authors gratefully acknowledge to Nara Institute of Science and Technology (NAIST), Nara, Japan to conduct this research using all kinds of facilities of Dependable System Laboratory. This work is not funded by any source.

## Author Contributions

**Conceptualization:** Fakir Sharif Hossain.

**Data curation:** Fakir Sharif Hossain, Rian Ferdian.

**Formal analysis:** Fakir Sharif Hossain.

**Investigation:** Fakir Sharif Hossain, Taiyeb Hasan Sakib.

**Methodology:** Fakir Sharif Hossain, Taiyeb Hasan Sakib, Rian Ferdian.

**Resources:** Muhammad Ashar.

**Software:** Fakir Sharif Hossain, Rian Ferdian.

**Validation:** Rian Ferdian.

**Visualization:** Fakir Sharif Hossain, Muhammad Ashar.

**Writing – original draft:** Fakir Sharif Hossain.

**Writing – review & editing:** Taiyeb Hasan Sakib, Muhammad Ashar, Rian Ferdian.

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
