## [Decision Letter · Decision Letter 0]

29 Sep 2021

PONE-D-21-17578A Dual Mode Self-Test for a Stand Alone AES CorePLOS ONE

Dear Dr. Hossain,

Thank you for submitting your manuscript to PLOS ONE. After careful consideration, we feel that it has merit but does not fully meet PLOS ONE’s publication criteria as it currently stands. Therefore, we invite you to submit a revised version of the manuscript that addresses the points raised during the review process. The manuscript had been reviewed by 2 reviewers. Reviewer 1 was of the view that manuscript describes technically sound piece of work while Reviewer 2 was of the opinion that manuscript partly describes the technically sound piece of work. Both reviewer had made certain comments and suggestions to improve the work. After thorough consideration of comments of both reviewers, my decision is "major revision". Please incorporate the suggestions/comments of both reviewers.

We look forward to receiving your revised manuscript.

Kind regards,

Gulistan Raja

Academic Editor

PLOS ONE

Additional Editor Comments (if provided):

Reviewers' comments:

Reviewer's Responses to Questions

**Comments to the Author**

1. Is the manuscript technically sound, and do the data support the conclusions?

Reviewer #1: Yes

Reviewer #2: Partly

2. Has the statistical analysis been performed appropriately and rigorously? 

Reviewer #1: Yes

Reviewer #2: Yes

3. Have the authors made all data underlying the findings in their manuscript fully available?

Reviewer #1: Yes

Reviewer #2: Yes

4. Is the manuscript presented in an intelligible fashion and written in standard English?

Reviewer #1: Yes

Reviewer #2: Yes

5. Review Comments to the Author

Reviewer #1: 1. Does the paper demonstrate an adequate understanding of relevant literature in the field and cite an appropriate range of literature sources? It seems that a few related works are included in the paper. More recently published papers should be discussed in related work section, and also be compared to the proposed method.

2. Is the paper's argument built on an appropriate base of theory, concepts or other ideas? Has the research or equivalent intellectual work on which the paper is based been well designed? How authors justify that the methods employed are appropriate?

3.Introduction you have to tell what is need to this research work? What you can get from this? Has any research work, which did this earlier? What is the motivation behind this research work? Please rewrite contributions, it is not describing the paper in the current form.

4.he discussion of Experiment should be written more in-depth, more precise and concrete, such as what questions were resolved? How can the proposed method solve these problems? The most recent works should be discussed in the related work section:

Enforcing position-based confidentiality with machine learning paradigm through mobile edge computing in real-time industrial informatics.

Energy-aware Geographic Routing for Real Time Workforce Monitoring in Industrial Informatics

Energy-Aware Green Adversary Model for Cyber Physical Security in Industrial System

5.The evaluation part is not detailed enough and lacks the description of the experimental settings. Furthermore, it becomes hard to understand the advantages of the proposed mechanism due to the lack of comparison with the existing schemes. Although authors have provided various comparative results, however, more details on how proposed phenomenon performs better results against baseline is still missing in the paper.

6.Authors should add a section which describe the flowchart shows the implementation procedure of the software.

7.Finally, it is suggested to improve writing of this manuscript to ameliorate this paper readability.

Reviewer #2: The authors proposed a novel self-test scheme aimed to the faults and Hardware Trojans on AES IP core. The experiment results show that it has high coverage and low overhead. However, there are some questions needed to be answered or discussed.

1. Is the proposed Dual mode Self-test structure limited to AES design? Providing that it is applied to other designs, the performance and the overhead should be introduced.

2. In Results section, only two types of Trojans in Trust-Hub are inserted. Although these two Trojans are relatively typical, it may be not enough to evaluate the performance on HT detection.

3. In Table 3, the area overhead of the proposed method is 0.091%, If scan chain buffers are included and scan mode is full-scan, the overhead should be more than that. The scan chain and clock gating circuit should be demonstrated in detail.

4. The resolution of Figure 2 and Figure 5 is not high enough.

6. PLOS authors have the option to publish the peer review history of their article (what does this mean?). If published, this will include your full peer review and any attached files.

Reviewer #1: **Yes: **Arun Kumar Sangaiah

Reviewer #2: No

---

## [Author Response · Author response to Decision Letter 0]

18 Oct 2021

Response to Reviewers

Editor’s Comments

Answer: Thank you so much for your comments. We have double check the manuscript for style requirements and followed the PlOS ONE style template. 

Answer: We have checked the manuscript language usage thoroughly and the proof reading is done by the following personnel.

Dr. Md. Altaf-Ul-Amin, Associate Prof., Nara Institute of Science and Technology (NAIST), Ikoma, Nara, Japan.

Review Comments to the Author

Reviewer #1: 

1. Does the paper demonstrate an adequate understanding of relevant literature in the field and cite an appropriate range of literature sources? It seems that a few related works are included in the paper. More recently published papers should be discussed in related work section, and also be compared to the proposed method.

2. 

Answer: Thank you so much for your comments. We have added 18 more recent and relevant articles in the Introduction, Related work and comparison sections so that our claim can be justified more clearly. References [4-18], [66-67] and [74] are added. Reference [74] is added to the comparison Table-4 in Subsection-4.4.

3. Is the paper's argument built on an appropriate base of theory, concepts or other ideas? Has the research or equivalent intellectual work on which the paper is based been well designed? How authors justify that the methods employed are appropriate?

Answer: Thank you so much for no comments.

4. Introduction you have to tell what is need to this research work? What you can get from this? Has any research work, which did this earlier? What is the motivation behind this research work? Please rewrite contributions, it is not describing the paper in the current form.

Answer: Thank you so much for your comments. We have rewritten the whole Introduction Section with 15 more relevant articles and added one more paragraph (from line no. 9 to 33) in this Section to display the contribution of our work. The added paragraph is like below.

Due to this impressive security potentiality of AES, it is being used in various emerging applications, either in software or hardware implementations. Hardware implementation of the algorithm offers higher security and speed than that of its software implementation. Due to enormous speed and security performances, now a lot of research for hardware realization of the AES cryptoprocessor is reported in the literature [4-13]. Some of the research focuses on hardware resource optimization [4-6], while some other on speed optimization [7-9] and some other on power consumption optimization [10-13]. A very few works on built-in-self-test (BIST) has reported in literature [14-18]. Some of them focus on-chip test pattern generation on detecting circuit aging [14-16], and some other structures detect Trojans [17, 18]. The testability and hardware Trojans are two major concerns that make the AES chip complex and vulnerable. The problem of testability in the complex AES chip is not addressed yet, and also, the hardware Trojan is a significant security threat that can leak secret key information easily if the hardware is compromised. Hardware Trojans are the manipulation or insertion of some extra transistors in a chip which can result in information leaking from the AES chip [19]. The research motivation of this work is to facilitate the BIST implementation in AES cryptography processors in terms of testability and hardware security domain. From a testability perspective, the on-chip BIST structure can significantly reduce the test cost with extra hardware and performance overhead. There is a great concern of hardware-based attacks from the security domain, like hardware Trojans in recent state-of-the-art. The AES chip is vulnerable to hardware Trojan attacks, as some significant research displayed. Therefore, the main purpose of this research is to address these problems by implementing a dual-mode BIST technique into the AES chip to address the testability issues and analyze the overhead of area, delay, and power.

The key points are below.

The research motivation of our work is to facilitate the BIST implementation in AES cryptography processor in terms of testability and hardware security domain. AES is a proven symmetric key and latest cryptography algorithm adopted by USA military. It is a proven fact that AES outperforms all other existing symmetric key cryptography algorithms. Therefore, we would like to introduce the concept of a dual-mode BIST architecture in designing the AES crypto-processor ASIC which is not reported in the literature yet. At today’s VLSI design, the testability and hardware security of a complex VLSI chip are the prime concern. In testability perspective, the focus of this research is to address the testability issues of the AES crypto-processor chip which is not reported in the literature to the best of our knowledge. The objective of any BIST technique is getting higher number of fault-coverage using lower number of test vectors. In security domain, it is a proven fact that different software based crypt-analytical attacks such as Brute-force, Linear crypt-analysis and Differential crypt-analysis, etc., have been proven ineffective to break the AES. However, there is a great concern of hardware based attack like hardware Trojans in resent state of the art. The AES chip is vulnerable at hardware Trojan attack as some significant research displayed. Therefore, the main purpose of this research is to address these problems by implementing a dual-mode BIST technique into the chip which can solve the testability issue and Trojan security treats. 

5. The discussion of Experiment should be written more in-depth, more precise and concrete, such as what questions were resolved? How can the proposed method solve these problems? The most recent works should be discussed in the related work section:

Enforcing position-based confidentiality with machine learning paradigm through mobile edge computing in real-time industrial informatics.

Energy-aware Geographic Routing for Real Time Workforce Monitoring in Industrial Informatics

Energy-Aware Green Adversary Model for Cyber Physical Security in Industrial System

Answer: Thank you for the comment. We would like to mention politely that our method is testing manufacturing faults and detecting hardware Trojans in the AES chip using built-in-self-test (BIST). This method is based on an on-chip testing of the AES chip under mixed mode (on-chip pattern storing and maximizing fault coverage) which does not reported in the literature yet, so far our knowledge. To facilitate our claim we perform design-for-testability with some hardware overhead and power consumption and compared to the state of the art. We would like to say politely that the mentioned titles are out of scope in this research. We have added 18 more articles to motivate the readers. We have added 1 more recent article [74] in the discussion and comparison Section 4.4.

6. The evaluation part is not detailed enough and lacks the description of the experimental settings. Furthermore, it becomes hard to understand the advantages of the proposed mechanism due to the lack of comparison with the existing schemes. Although authors have provided various comparative results, however, more details on how proposed phenomenon performs better results against baseline is still missing in the paper.

Answer: Thank you so much for the comment. We have added 18 more relevant papers and provided Figure-3 (Fig 3. Flowchart representation of the proposed PEAB pair generation) for getting PEAB pairs in our proposed DMST method. We have added Fig.6 to display the current sensor position in the layout of the AES chip. Fig.3, Fig. 6 and Fig.7 may help to understand the experimental setting and evaluation of the proposed method. A detailed evaluation results is presented in a new added Table-2 (see below). One more recent article is added in the comparison Table-4. Subsection-4.4 (Discussion and Comparison) is rewritten in details. 

Table-2

Event Complete

 design Scan 

design Clock 

gating Memory ADC Comparator

Total Area

 in um width 2163275.57 110275.2 1968.58 520 241.34 163.18

Area Overhead 5.239% 5.097% 0.091% 0.024% 0.011% 0.0075%

Average power 

in mW per block 17.521 3.627 0.0647 0.0171 0.007938 0.005368

Power overhead 

 per block 21.073% 20.703% 0.369% 0.098% 0.045% 0.031%

Critical path 

delay in ps 592 178.38 50.0786 13.228 6.139 4.151

Delay overhead 42.56% 30.132% 8.459% 2.234% 1.037% 0.701%

7. Authors should add a section which describe the flowchart shows the implementation procedure of the software.

Answer: Thank you so much for the comment. Subsection 3.2 (Overview) describe of getting our proposed Power equal adjacent block (PEAB) pairs which is also displayed in Fig.3 as flow chart. The required hardware is described in Section 3.6 with the help of Fig. 6 (newly added) and Fig. 7. We have rewritten Section 3.6 in details.

8. Finally, it is suggested to improve writing of this manuscript to ameliorate this paper readability.

Answer: We have checked the manuscript language usage thoroughly and proof reading is done by the following personnel. Dr. Md. Altaf-Ul-Amin, Associate Prof., NAIST, Nara, Japan.

Reviewer #2: 

1. Is the proposed Dual mode Self-test structure limited to AES design? Providing that it is applied to other designs, the performance and the overhead should be introduced.

Answer: Thank you so much for your valuable comment. Our proposed dual mode self-test structure is not limited to only AES design. This method is applicable to other ASIC designs. However, the research motivation of our work is to facilitate the BIST implementation in AES cryptography processor in terms of testability and hardware security domain. To the best of our knowledge, the testability issue of AES in hardware domain is not addressed yet. All the attempts are performed in software domain. We solve the testability issue by built-in-self-test (BIST) of AES chip in hardware domain. We propose a dual mode BIST structure that tests the AES chip with on-chip hardware. AES is a proven symmetric key and latest cryptography algorithm adopted by USA military. It is a proven fact that AES outperforms all other existing symmetric key cryptography algorithms. Therefore, we would like to introduce the concept of a dual-mode BIST architecture in designing the AES crypto-processor ASIC which is not reported in the literature yet. At today’s VLSI design, the testability and hardware security of a complex VLSI chip are the prime concern. In testability perspective, the focus of this research is to address the testability issues of the AES crypto-processor chip which is not reported in the literature to the best of our knowledge. The objective of any BIST technique is getting higher number of fault-coverage using lower number of test vectors. In security domain, it is a proven fact that different software based crypt-analytical attacks such as Brute-force, Linear crypt-analysis and Differential crypt-analysis, etc., have been proven ineffective to break the AES. However, there is a great concern of hardware based attack like hardware Trojans in resent state of the art. The AES chip is vulnerable at hardware Trojan attack as some significant research displayed. The main purpose of this research is to address these problems by implementing a dual-mode BIST technique into the chip which can solve the testability issue and the Trojan security treats. 

The Introduction section is rewritten thoroughly to display the motivation of our work and added 18 more recent and relevant articles. We have added Table-2 to display more details of our proposed DMST method.

2. In Results section, only two types of Trojans in Trust-Hub are inserted. Although these two Trojans are relatively typical, it may be not enough to evaluate the performance on HT detection.

Answer: Thank you so much for your comment. We incorporated 2 more sequential Trojans from Trust Hub to validate our claim. AES-T300 and AES-T400 are two sequential Trojans that leak information when triggered. We have inserted them without any modifications.

3. In Table 3, the area overhead of the proposed method is 0.091%, If scan chain buffers are included and scan mode is full-scan, the overhead should be more than that. The scan chain and clock gating circuit should be demonstrated in detail.

Answer: Thank you so much for your comment. Our motivation is that, the scan chain technology is a well established testing method and people in this field consider the extra hardware cost as design-for-testability purposes. Therefore, the scan chain overhead is not considered in our technique rather we consider it as already included in the design. However, the test controller, the clock gating controller, memory, embedded current sensors and comparator unit are the extra hardware to obtain the power side-channel parameters. We have added Table-2, in which a detailed overhead of the test controller, clock gating controller, 16 current sensors, comparator circuit and memory units is presented in the result section. Fig 6 in Subsection-3.6 is added to display the current sensors placement. We have rewritten Subsection 3.6 to demonstrate our proposed test controller unit.

Table-2

Event Complete

 design Scan 

design Clock 

gating Memory ADC Comparator

Total Area

 in um width 2163275.57 110275.2 1968.58 520 241.34 163.18

Area Overhead 5.239% 5.097% 0.091% 0.024% 0.011% 0.0075%

Average power 

in mW per block 17.521 3.627 0.0647 0.0171 0.007938 0.005368

Power overhead 

 per block 21.073% 20.703% 0.369% 0.098% 0.045% 0.031%

Critical path 

delay in ps 592 178.38 50.0786 13.228 6.139 4.151

Delay overhead 42.56% 30.132% 8.459% 2.234% 1.037% 0.701%

4. The resolution of Figure 2 and Figure 5 is not high enough.

Answer: Thank you so much for your comments. We have redrawn Figure-2 and Figure-5 with high resolution as per Plos One standard of 300dpi.

---

## [Decision Letter · Decision Letter 1]

2 Dec 2021

A Dual Mode Self-Test for a Stand Alone AES Core

PONE-D-21-17578R1

Dear Dr. Hossain,

We’re pleased to inform you that your manuscript has been judged scientifically suitable for publication and will be formally accepted for publication once it meets all outstanding technical requirements.

Kind regards,

Gulistan Raja

Academic Editor

PLOS ONE

Additional Editor Comments (optional):

Reviewers' comments:

Reviewer's Responses to Questions

**Comments to the Author**

1. If the authors have adequately addressed your comments raised in a previous round of review and you feel that this manuscript is now acceptable for publication, you may indicate that here to bypass the “Comments to the Author” section, enter your conflict of interest statement in the “Confidential to Editor” section, and submit your "Accept" recommendation.

Reviewer #1: All comments have been addressed

Reviewer #2: All comments have been addressed

2. Is the manuscript technically sound, and do the data support the conclusions?

Reviewer #1: Yes

Reviewer #2: Yes

3. Has the statistical analysis been performed appropriately and rigorously? 

Reviewer #1: Yes

Reviewer #2: Yes

4. Have the authors made all data underlying the findings in their manuscript fully available?

Reviewer #1: Yes

Reviewer #2: Yes

5. Is the manuscript presented in an intelligible fashion and written in standard English?

Reviewer #1: Yes

Reviewer #2: Yes

6. Review Comments to the Author

Reviewer #1: (No Response)

Reviewer #2: The comments are answered in detail, the related references are added and the resolution of figures is improved.

In my opinion, this paper can be considered for publication.

7. PLOS authors have the option to publish the peer review history of their article (what does this mean?). If published, this will include your full peer review and any attached files.

Reviewer #1: No

Reviewer #2: No

---

## [Editor Report · Acceptance letter]

10 Dec 2021

PONE-D-21-17578R1 

A Dual Mode Self-Test for a Stand Alone AES Core 

Dear Dr. Hossain:

I'm pleased to inform you that your manuscript has been deemed suitable for publication in PLOS ONE. Congratulations! Your manuscript is now with our production department. 

Kind regards, 

on behalf of

Dr. Gulistan Raja 

Academic Editor

PLOS ONE